# Risk Factors Associated with Post-Operative Complications in Multidisciplinary Treatment of Descending Necrotizing Mediastinitis

**DOI:** 10.3390/jcm11216364

**Published:** 2022-10-28

**Authors:** Maria Teresa Congedo, Dania Nachira, Mariano Alberto Pennisi, Marco Chiappetta, Giuseppe Calabrese, Giuseppe Bello, Claudio Parrilla, Laura Franza, Marcello Covino, Leonardo Petracca Ciavarella, Venanzio Porziella, Maria Letizia Vita, Filippo Lococo, Stefano Margaritora, Elisa Meacci

**Affiliations:** 1Unit of Thoracic Surgery, Fondazione Policlinico Universitario A. Gemelli IRCCS, 00168 Rome, Italy; 2Facoltà di Medicina e Chirurgia, Università Cattolica del Sacro Cuore, 00168 Rome, Italy; 3Department of Intensive Care Medicine and Anesthesiology, Fondazione Policlinico Universitario A. Gemelli IRCCS, 00168 Rome, Italy; 4Unit of Otorhinolaryngology, Head and Neck Department, Fondazione Policlinico Universitario A. Gemelli IRCCS, 00168 Rome, Italy; 5Emergency Department, Fondazione Policlinico Universitario A. Gemelli IRCCS, 00168 Rome, Italy

**Keywords:** descending necrotizing mediastinitis, cervical abscess, surgery, thoracoscopy

## Abstract

Background: Descending necrotizing mediastinitis (DNM) is a severe, life-threatening complication of oropharyngeal infections with cervical necrotizing fasciitis. In this study, we aimed to identify any possible factors that correlate with favorable outcomes. Methods: We retrospectively analyzed our series of 18 patients who underwent surgical treatment for DNM from a cervical abscess. Gender, age, symptoms, etiopathogenesis, comorbidities, time to surgery from diagnosis, degree of diffusion, identified microorganisms, surgical procedure, days in the intensive care unit, need for tracheostomy, complications, and surgical outcomes were reviewed. Results: The main type of surgery was thoracotomy + cervicotomy in eight cases (50.0%), followed by cervicotomy +VATS in four (22.2%). Seven patients (38.9%) had two or more surgeries; a bilateral operation was necessary for four patients. Evaluating the risk factors associated with post-operative complications, age ≥ 60 years (*p*:0.031), cervicotomy alone as surgical approach (*p* = 0.040), and the bilateral approach (*p* = 0.048) resulted in significance in terms of the univariate analysis; age ≥ 60 years (*p* = 0.04) and cervical approach (*p* = 0.05) maintained their significance in terms of the multivariate analysis. Conclusions: The low mortality of our series emphasizes the importance of an extensive and immediate surgical drainage of both the neck and the mediastinum. Mediastinal drainage from cervicotomy seems to be a risk factor for post-operative complications. Minimally invasive surgery on the chest cavity, such as with Uniportal-VATS, could be a good approach above all in elderly patients and all those cases where bilateral access is required.

## 1. Introduction

Descending necrotizing mediastinitis (DNM) is a dangerous and life-threatening form of soft tissue infection with a high mortality rate, even in surgically treated patients. Mediastinitis can usually originate from endoscopic or foreign body perforation, as well as from surgical wounds, in particular post-cardiac surgery and cancer. In some cases, DNM can develop from an odontogenic, oropharyngeal, or cervical abscess; in these cases, it spreads from the soft tissue through the deep and superficial cervical fascial planes to the mediastinum and it disseminates, following gravity and negative pressure, into the chest cavity. Infections from the pharynx, the tonsils, or teeth can affect the retropharyngeal paratracheal, paraoesophageal, prevertebral, posterior mediastinum, and pleural spaces [1]. Delay in the diagnosis of this condition may lead to rapid progression of the disease and fatal outcomes; indeed, if not promptly diagnosed and operated upon, the mortality remains very high, around 60–70% [2,3]. Involved patients often present immunosuppression or systemic diseases (e.g., diabetes, HIV, malignancies requiring chemotherapy or immunotherapy and immunologic diseases) that predispose them to this pathology.

All the series published since 1970 [4,5] confirmed that cervical drainage alone is often insufficient to control the infection, and mortality rate from drainage alone is significantly higher than mediastinal debridement [6]. On the basis of the experience accumulated over the last few years, we aimed to evaluate the results of aggressive, prompt, and multidisciplinary management of DNM at our institution and identify any possible factors that correlate with favorable outcomes.

## 2. Materials and Methods

We retrospectively analyzed our series of 18 patients who were admitted to the emergency room (E.R.) for DNM from a cervical abscess and underwent prompt surgical treatment from January 2012 to February 2022. Patients with acute mediastinitis secondary to esophageal perforation, surgical wound infection, or anastomotic leak were excluded.

Gender, age, symptoms, etiopathogenesis, comorbidities (diabetes, immunodeficiency, cancer), time to surgery from diagnosis, degree of diffusion, identified microorganisms, surgical procedure, days in an intensive care unit (ICU), need for tracheostomy, complications, and surgical outcomes were reviewed.

### 2.1. Preoperative Assessment

All patients underwent blood tests, electrocardiography, and neck and chest computed tomography (CT). Surgery was performed under general anesthesia and single lung ventilation in the case of thoracic access. Two patients were excluded from surgery for hemodynamic instability and died of septic shock after a few hours.

In the suspicion of DNM, the patient promptly underwent a neck and chest CT scan with intravenous iodinated contrast media. In 4 cases, because of the suspicion of pharyngoesophageal perforation, a gastrographin swallow was also performed.

### 2.2. Surgical Intervention

Cervical access was an extended “J” shape access along the sternocleidomastoid muscle of the involved side. In the case of bilateral involvement of the cervical compartments, an extended Kocher cervicotomy was performed. At the end of the surgery, drainages were left in place in the cervical compartments of the neck and retrosternal region through the cervical incision. The skin and subcutaneous muscles were often not closed to allow for further daily medications of the neck.

The anterior and posterior mediastinum and chest cavity were drained through a V intercostal access or by thoracotomy or Uniportal Video-Assisted Thoracic surgery (VATS). Two (in Uniportal-VATS) or three (in thoracotomy) drainages were left in place in the mediastinum and pleural space.

### 2.3. Statistical Analysis

Continuous variables were expressed as mean ± standard deviation, and categorical ones as percentage. Pearson’s χ^2^ test and Fisher’s exact test were used to compare categorical variables, and Student’s *t*-test was used to compare continuous variables.

All the clinical-pathological and surgical variables were tested in univariate analysis to assess prognostic factors. All covariates with *p* < 0.2 were selected for the Cox proportional hazards regression model to assess prognostic independent factors. For all tests, a *p*-value < 0.05 was considered statistically significant. Statistical analysis was performed using IBM SPSS Statistics for Macintosh (version 25.0, IBM Corp., Armonk, NY, USA).

## 3. Results

Among the studied population, seven (38.9%) patients were males. The median age was 43 years for males (range 23–58 years) and 63 for females (range 41–91 years). Five patients (27.8%) were over 60 years old. Seven patients were referred to our hospital from another hospital with the suspicion of DNM. Etiopathogenesis was odontogenic, peritonsillar, and pharyngeal in nine, seven, and four cases, respectively. The main clinical-pathological characteristics, comorbidities, and symptoms of the patients are reported in Table 1.

Out of the 18 patients, 17 underwent surgery during the first 12 h from admission to the E.R. and one on the first day. One patient was operated after 6 days from the first admission in another hospital: she was a 48 years old woman who had a dental extraction ten days before and that noticed cervical swelling and odynophagia (Figure 1). After one week of oral antibiotic intake without benefit, she was admitted to a general hospital for fever and cervical pain. An ultrasound showed only edema of the cervical fascial layers; for this reason, she started intravenous antibiotic therapy, but after 6 days, she had spontaneous drainage of pus from the neck. Then she was referred to our hospital and immediately operated (she underwent two cervicotomies and three right thoracotomies, along with vacuum-assisted closure therapy because of cervico-thoracic deep infection).

Seven patients were referred to our hospital from another institution for further management and immediately underwent surgery. Usually, they experienced a failure of antibiotic therapy, and for worsening clinical conditions, they performed a CT scan showing a DNM, wherein the individual was immediately transferred to our Emergency Department.

The main type of surgical approach as first operation was thoracotomy + cervicotomy in eight cases (50.0%), followed by cervicotomy + VATS in four cases (22.2%). The right side was the most affected side (11, 61.1%). Seven patients (38.9%) underwent two or more surgeries, for a mean number of re-operations of 2.5 ± 2.7 (1–11); a bilateral operation was necessary for four of the patients. None of the patients who underwent Uniportal-VATS required a second look on the homolateral hemithorax, but two patients required a Uniportal-VATS on the contralateral side.

In three cases (16.7%), only a cervicotomy with the insertion of retrosternal drainage in the superior mediastinum was performed without opening the chest cavity.

The first was a 36-year-old male who had a left third mandibular molar odontectomy 5 days before the evidence of cheek swelling and pain. He went to another hospital where he was discharged with a prescription for antibiotics and corticosteroids. The day after, he went to our Emergency Department with a voluminous left face swelling and subcutaneous crackling with fever and confusion (Figure 2). After neck and thorax CT scan were performed, the patient was transferred to the operating room for drainage of cervical abscess and insertion of two transcervical drainages, one into the anterior and one into the posterior mediastinum. The cervical wound was kept open and packed to permit daily irrigation with iodopovidone and saline solution. One week later, for worsening clinical conditions, with the onset of pleural and pericardial effusion, the patient underwent subxiphoid pericardial drainage, right thoracotomy and insertion of three pleural tubes, and a second cervicotomy. Seven days later, the patient went again to the operating room for a second right thoracotomy and left thoracoscopy for the onset of purulent fluid collections in the aorto-pulmonary window, close to the subclavian artery and along the left pericardium. After this bilateral operation, the patient’s conditions improved, and he was discharged on postoperative day 36 without sequelae.

In the other cases, the choice of performing mediastinal drainage through the cervical incision was due to the age (85 years old) of the patient in the second case and due to comorbidities (diabetes and autoimmune thrombocytopenia) in the third case. In these cases, DNM was caused by a tonsillar infection and a pharyngeal abscess spreading in the superior mediastinum and anterior mediastinum (type Ia and I, respectively, according to Guan’s classification).

All patients required ICU stay, and the time of intubation was 13.9 ± 11.4 (1–41) days. In three cases, a tracheostomy was performed (one for mucus retention and inefficient cough, and two for otolaryngologist’s choice to protect the airway during cervicotomy, in one case for Ludwig’s angina onset).

Eight patients (44.4%) developed post-operative complications (ventilation-acquired pneumonia, pulmonary embolism, delirium, and CRYMINE; see Figure 3).

Two (11.1%) patients died. The first was a diabetic 63-year-old woman who had an arthrodesis for scoliosis 2 years before the DNM from pharyngeal abscess. Despite the cervicotomy and the posterior mediastinal incision in the left thoracotomy and the removal of the infected vertebral pedicle screw, she died of sepsis in the ICU 20 days after thoracic surgery.

The second patient was a 91-year-old woman, referred to our hospital for dyspnea in submandibular, cervical, and mediastinal fluid collection from a dental abscess. She underwent cervicotomy and right thoracotomy, but she died of respiratory and renal failure 14 days after surgery.

All patients received empiric broad-spectrum antimicrobial therapy including antifungal agents until the lab reports of the intraoperative microbiological examinations became available. Then, targeted therapy began, according to microbiologic results.

The main isolated germs were as follows: Streptococci in five cases (sanguis, anginosus, epidermidis, constellatus, and Peptostreptococcus anaerobius), Prevotella (melaninogenica, baroniae, and spp.) in three cases, Lactobacillus casei and paracasei in two cases, and Candida albicans in two cases; other bacteria were Klebsiella pneumoniae, Escherichia coli, Enterococcus fecalis, and Acinetobacter baumanii. In four cases, polymicrobial infections resulted in the cultures. Two patients were negative in the microbiological tests.

The total length of in-hospital stay was 31.2 ± 25.9 (9–101) days.

Demographic, clinical features and surgical outcomes of patients are listened in Appendix A.

No correlation was found between clinical and surgical variables (gender, age, comorbidities, type of surgery, side, re-operations) and total in-hospital stay.

Evaluating the risk factors associated with post-operative complications, among age >=60 years (*p* = 0.031), only cervicotomy alone as a surgical approach (*p* = 0.040) and bilateral approach (*p* = 0.048) resulted in being significant at univariate analysis, and only age >=60 years (*p* = 0.04) and cervical approach (*p* = 0.05) confirmed their significance in terms of the multivariate analysis (see Table 2).

## 4. Discussion

Estrere et al. [7], at the beginning of the 1980s, first described DNM in 10 patients with mediastinitis complicating an oropharyngeal infection.

In 1999, Endo [8] published the guidelines for surgical management based on diffusion of DNM in four patients, and he distinguished three different types of DNM: type I, including superior mediastinal infection above the carina; type IIA, including superior and antero-inferior infection; and type II B, including total mediastinal infection.

In 2021, from the analysis of 139 patients with DNM caused by odontogenic or pharyngeal infection, a new classification was proposed by Guan [9], which was based on the progression of mediastinal infection. It includes the DNM of posterior mediastinum, usually originating from retropharyngeal space abscesses descending through the prevertebral space.

This new classification includes four types of DNM: type Ia, in which infection is confined in the antero-superior mediastinum; type I, in which infection involves the entire anterior mediastinum; type II, in which infection involves the postero-superior mediastinum or the posterior mediastinum; and type III, in which infection involves both the anterior and posterior mediastinum.

In type Ia, patients should be managed by only transcervical mediastinal drainage; instead, patients of types I and II DNM require a thoracic approach (VATS or open). The patients with type III were managed with unilateral or bilateral thoracic surgery, recording the highest mortality in the whole group (27.6% vs. 9.4%).

In the literature, several papers suggested cervicotomy for limited DNM. In a German series of 45 DNM [10], formal thoracotomy was reserved for cases extending below the plane of the tracheal bifurcation, since a favorable outcome was recorded in more than 80% of patients. The authors suggested an early transcervical mediastinal approach in DNM limited to the upper mediastinum with close follow-up.

Moreover, in a recent work including nine patients [11], a transcervical approach for deep cervical and paratracheal fluid collections was preferred to control all types of DNM because it was considered less invasive than transthoracic approaches via thoracotomy; thoracotomy was reserved, as a second-step surgery, to patients whose infection was not controlled by cervical drainage.

In 2010, Karkas et al. [12] published an algorithm of DNM surgical management that was based on their experience with 17 patients. All patients underwent cervicotomy; in 10 patients with subcarinal involvement (type I according to Endo’s classification) [8], they used a pretracheal approach for anterosuperior mediastinum and a retro visceral approach for posterosuperior mediastinum; in seven patients with infracarinal involvement (type II), the antero-inferior mediastinum was approached by sternotomy and postero-inferior mediastinum by posterolateral thoracotomy.

On the contrary, in a recent work, De Palma et al. [13] recommended a lateral thoracotomy in addition to cervicotomy also in patients with initially limited mediastinitis, with the aim of obtaining a toilette of pleural collections and a mediastinal debridement (after opening the mediastinal pleura).

Our results confirmed that the strategy suggested by De Palma et al. [13] is effective: only transcervical drainage of the mediastinum by cervicotomy (performed in three patients) seemed to be a risk factor for postoperative complications in our series.

In the last 6 years, the VATS approach and, in particular, Uniportal-VATS has become the preferred technique for all major thoracic procedures in our department [14]; therefore, urgent mediastinal and pleural toilette in cases of DNM started to be performed through this approach.

According to more recent literature [15], we consider VATS to be more effective for the management of DNM because it allows for a good toilette of the mediastinum and chest cavity while guaranteeing a minimally invasiveness. Uniportal-VATS is often better than a thoracotomy in providing good vision and exposure in the hands of expert surgeons; it allows for the opening of the mediastinal pleura from the superior thoracic inlet to azygos vein on the right side and, if necessary, for the ability to proceed posteriorly along the paraoesophageal and subcarinal spaces. Pus aspiration and necrotic debridement of all the foci described in the CT scan are possible under direct vision through a 4 cm incision at a V intercostal space.

Compared to mediastinal drainage from cervicotomy, VATS has the advantage of exploring the entire chest cavity and to place the drainages under visualization into the pleural collections with a very low traumatic impairment. At the same time, since the right chest cavity is usually firstly involved in the DNM, in the case of contralateral spread, a Uniportal-VATS approach is feasible with all the advantages of lower invasiveness.

Three out of the four patients requiring contralateral lung surgery had post-operative complications (pneumonia, delirium, and pericardial effusion). A bilateral approach was necessary for two patients with stage II according to Guan’s classification, one with stage I one with stage III. In no case did we perform a bilateral approach during the first operation. All the patients had a first operation on the right side and a second operation on the left side. This means that the CT scan performed after the operation showed a progression of the disease and an involvement of the controlateral hemithorax. In our results, a bilateral approach, even if only at univariate analysis, is a risk factor for post-operative complications, due to both the worsening of the mediastinal involvement and the impact of the functional respiratory impairment, which consequently entails a longer hospitalization in the ICU.

In the last years, we switched from a more conservative surgical policy (only cervicostomy) to a more aggressive one, despite being performed through mini-invasive Uniportal thoracoscopic approaches; this approach could also be offered to elderly patients in a situation of intensive multiparametric management because these patients are at high risk of major complications or death.

However, some authors [16,17] preferred sternotomy or clamshell incision for DNM because they ensure good access to the anterior mediastinum and to both thoracic cavities; in a single-center series published in 2012 [18], 16 patients with diffuse DNM were surgically treated by median sternotomy (*n* = 8) or the clamshell (*n* = 8) approach, with a mortality rate of 12.5% (2/16; the first for MOF 48 h after clamshell incision and the second for ARDS 4 months later, after several redo-clamshell operations). Although the authors did not correlate the cause of the exitus with the technical choice of surgery, we consider this approach at risk for osteomyelitis and dehiscence of the sternum; furthermore, it seems not so suitable for draining the infero-posterior mediastinum collections, even when performed in single-lung ventilation.

In this study, we attempted to identify clinical factors able to predict post-operative complications with the aim of suggesting to physicians which patients deserve more intensive care and multidisciplinary management (because of those more at risk for post-op complications). Older patients and patients treated by cervicostomy presented a significantly higher number of post-operative complications.

Despite the main limitations of our work, related to small sample size and the retrospective nature of the study, which prevent the analysis to drive strong conclusions, we report the results of a quite large series compared to other studies [11,12,13,16,17] in the relative short period of 10 years, considering the rarity of the disease. The main strengths of this study are the good results in terms of mortality (11.1%) and low rate of post-operative complications (44.4%), lower than the data reported in the literature [2,3] and similar to De Palma’s results (null mortality with 40% complications in a young population with a mean age of 49.07 ± 14.92 years [13]). These results are even more interesting when considering the age of our population (27.8% being over 60 years old), as this resulted in being a risk factor for complications

Our good results can be probably explained by a multidisciplinary, prompt, and aggressive management of the patient in the first hours by intensivists, thoracic surgeons, anesthesiologists, otolaryngologists, and dentists; another reason was the integrated treatments with our infectivologists in the post-operative period, which added the benefit of target antibiotic therapies to daily surgical medication of the neck through the cervicotomy kept open. All drainages were taken in place and washed with physiological solution until their tribute was clear without purulent material. This meticulous management added to the long recovery period of the illness itself, explaining the long hospitalizations of 31.2 ± 25.9 days (9–101).

Another point of strength in our results is the low rate of tracheotomy (16.9%) compared to the reported 60% of other series [13]. On this point, indeed, we agree with Kim [19], who considered tracheotomy not always necessary in DNM. In this Korean study [19] among seven (35%) patients with a tracheotomy, mortality rates of early tracheotomy and late tracheotomy were 75 and 0%, respectively, with early tracheotomy showing significantly higher mortality (*p* = 0.032). Although in our study tracheotomy was not a significant risk factor for mortality, one of the three patients who died in ICU underwent a tracheotomy. As in Kim’s series, in our report, patients with early tracheotomy experienced a prolonged hospital stay due to contamination of the pretracheal space requiring a flap for tracheo-cutaneous fistula closure. For this reason, we believe that tracheotomy should be performed with great caution in patients with DNM.

On the basis of multivariate analysis, we suggest avoiding mediastinal drainage from cervicotomy or reserving this approach for very limited antero-superior mediastinitis, since a non-invasive surgery could improve the survival in DNM patients, without impairing complications and long stay hospitalization.

## 5. Conclusions

DNM is a life-threatening condition that requires timely multidisciplinary management, with prompt and aggressive surgical treatment.

Standard treatment consists of cervicotomy to drain the source of infection, as well as an open approach to the thorax (either thoracotomy or sternotomy). Minimally invasive surgery on the chest cavity, such as in Uniportal-VATS in our experience, provides good exposure in order to clean the mediastinum and pleural cavity, with the advantages of reduced invasiveness, above all in elderly patients, and all those cases where bilateral access is required. This approach could also be offered to elderly patients in intensive multiparametric management because this group resulted a in having a high risk of major complications or death.

Further multicenter and randomized studies must take place in the future to drive stronger conclusions, identifying potential risk factors in clinical characteristics of patients or surgical approaches that may help clinicians in providing the most appropriate treatment for this quite rare but severe condition.

## Figures and Tables

**Figure 1 jcm-11-06364-f001:**
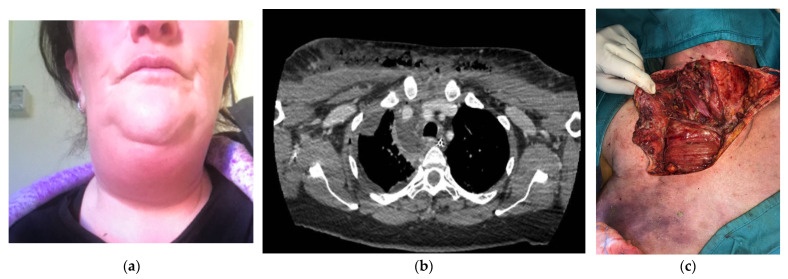
(**a**) A 48-year-old woman with cervical swelling and odynophagia. (**b**) A preoperative computed tomography scan showing the spread of infection extending to the lower anterior mediastinum, below the tracheal bifurcation with associated right pleural effusion. (**c**) Ample cervicotomy extended to the anterior pectoral exposition for a cervico-thoracic deep infection.

**Figure 2 jcm-11-06364-f002:**
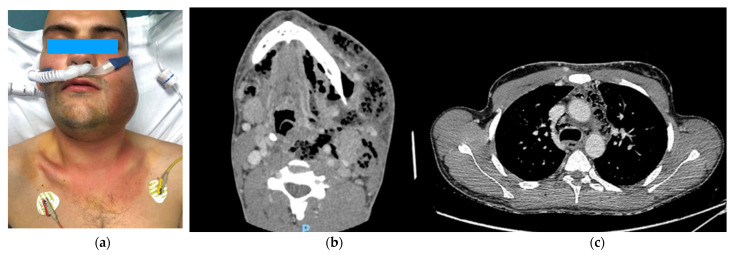
(**a**) A 36-year-old man with cheek swelling after tooth extraction. (**b**) A preoperative computed tomography scan showing a voluminous collection with hypodense material, multiple sepiments, and an extensive aerial component in the context at the epicenter in the left tonsillar loggia. (**c**) A mediastinal abscess with gas pocket formation in the anterior mediastinum and around the trachea.

**Figure 3 jcm-11-06364-f003:**
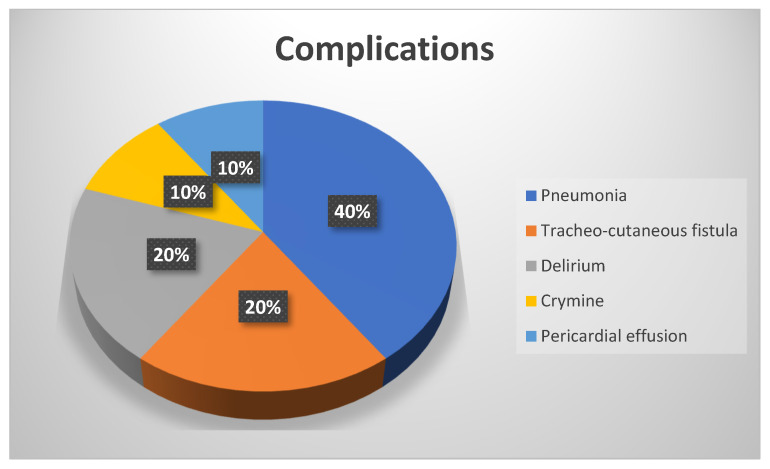
Postoperative complications.

**Table 1 jcm-11-06364-t001:** Clinical-pathological characteristics of the patients.

Variables	18 Patients
Age (≥60 years)	5 (27.8%)
Gender (male)	7 (38.9%)
COPD	1 (5.6%)
Smoking attitude	3 (16.7%)
Cardiovascular diseases	1 (5.6%)
Diabetes	6 (33.3%)
Symptoms at onset Swelling Fever Pharyngodinia Dyspnea	10 (55.6%) 6 (33.3%) 6 (33.3%) 5 (27.8%)
Guan’s classification 1 2 3	4 (22.2%) 6 (33.3%) 8 (44.5%)
Side Right Left Bilateral	11 (61.1%) 3 (16.7%) 4 (22.2%)
Type of first surgery Cervicotomy + thoracotomy Cervicotomy + VATS Cervicotomy alone (mediastinal drainage) Thoracotomy VATS	8 (50.0%) 4 (22.2%) 3 (16.7%) 1 (5.6%) 1 (5.6%)
Number of operations	2.5 ± 2.7 (1–11)
Tracheostomy	3 (16.9%)
Days of intubation	13.9 ± 11.4 (1–41)
In hospital stay (days)	31.2 ± 25.9 (9–101)
Complications	8 (44.4%)
Mortality	2 (11.1%)

**Table 2 jcm-11-06364-t002:** Risk factors for post-operative complications at univariate and multivariate analyses.

Variables	Univariate Analysis *p*-Value	Multivariate Analysis HR (95% CI) *p*-Value
Age ≥ 60 years	0.031	1.821 (0.793–2.245) *p* = 0.04
Surgical approach (only cervicotomy)	0.040	2.567 (1.789–3.972) *p* = 0.05
Bilateral approach	0.048	

## Data Availability

Data are the property of the Fondazione Policlinico Universitario A. Gemelli and may be visualized if needed after the authors’ approval.

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
