# Peer review of "Risk Factors Associated with Post-Operative Complications in Multidisciplinary Treatment of Descending Necrotizing Mediastinitis"

_jcm, 2022, doi:10.3390/jcm11216364_

Round 1

Reviewer 1 Report (Previous Reviewer 2)

What is the novelty of this study?

What is the significance of extracting risk factors of post-operative complications? No discussion has been given to this results.

Author Response

Reviewer 1:

What is the novelty of this study?

Reply: Thanks for your comment. Several studies have been reported investigating the early-term outcomes of mediastinitis (related to different clinical conditions) and the best therapeutic approach. On the other hand, there is not so robust data concerning the surgical approach of a specific population of patients affected by Descending Necrotizing Mediastinitis (DNM). In this study, we have tried to report the different strategy of care in our experience. Firstly, we observed that an “aggressive surgical treatment” (despite consisting of mini-invasive thoracoscopic (uni/bilateral) approach) was associated with better results when compared with cervicotomy alone.

Moreover, despite limited by the small number of cases, we have tried to identify clinical factors able to predict post-operative complications. The final “Take-home message” is to suggest a practical strategy of care based on “mini-invasive” but extended surgical approaches performed quickly after the diagnosis of DNM. At the same time, we wish to suggest to physicians which patients deserves an intensive care (because more at risk of post-op complication).

What is the significance of extracting risk factors of post-operative complications? No discussion has been given to this results

Reply: Thank you very much for your constructive suggestion. As reported before, we have tried to identify clinical factors able to predict post-operative complication with the aim of suggesting to physicians which patients deserve more intensive care and multidisciplinary management (because more at risk of post-op complication). Older patients and patients treated by cervicostomy only presented significantly higher number of post-operative complications.

As reported in the manuscript, in the last years, we switch from a more conservative surgical policy (only cervicostomy) to a more aggressive one (despite performed by mini-invasive uniportal thoracoscopic approaches); this approach could be offered also in elderly patients in an intensive multiparametric management because these patients resulted at high risk of major complications or death.

Following your suggestions, these observations were included in the discussion

Maria Teresa Congedo,

On the behalf of all co-Authors

Reviewer 2 Report (Previous Reviewer 1)

This manuscript was significantly improved after revision and the authors did great job in the discussion section. Although the material methods and results are still somewhat lengthy, I think this manuscript do bring new insight to the management of descending necrotizing mediastinitis. 

Author Response

This manuscript was significantly improved after revision and the authors did great job in the discussion section. Although the material methods and results are still somewhat lengthy, I think this manuscript do bring new insight to the management of descending necrotizing mediastinitis.

Reply: Thanks for your positive comment. We completely agree with You. The Materials and Methods were shortened in the revised version of the manuscript.

Maria Teresa Congedo,

On the behalf of all co-Authors

This manuscript is a resubmission of an earlier submission. The following is a list of the peer review reports and author responses from that submission.

Round 1

Reviewer 1 Report

Overall, this is a useful study in the literature regarding the descending necrotizing mediastinitis. However, lots of concern were raised and the paragraph in the discussion is chaos.

Line 26-29: This is somewhat ambiguous and may rephrase for better understanding.

Line 32 Uniportal or uniportal?

Line 38 and 39 should be in cointinous.

Line 44: interest?

Line 48-50: such as diabetes….

Line 53 and 54 should be in continuous.

Line 64-76 may present in the following “preoperative assessment”

Line 85-94 is the surgical intervention rather than “preoperative” assessment.

The typesetting of table 1 should be revised.

Line 133-135: This should be list in the “Material and methods”

Line 154-160, these sentences should be in continuity. All the description in the “reults” should be summarized. The authors presented clinical cases in separate parts of the results, which could be integrated.

The information in table 1 can be considered to be variables in the univariate and multivariate analyses.

The whole discussion should be summarized for more clear presentation.

Line 318: the conclusion is definitely too lengthy.

Author Response

Dear Reviewer,

thank you for your comments. We correct the manuscript following your suggestions. The  discussion and the conclusion were rewritten and summarized. 

I hope this version will obtain your consent and endorsement.

Kind regards

Maria Teresa Congedo on the behalf of the co-Authors.

Reviewer 2 Report

Thank you for this review oppotunity.

Dr. Congedo et al. analyzed 18 DNM patients and aimed to identify any possible factors that correlate with favorable outcomes.

They concluded that it was important to drain extensively and immediately for treatment of DNM. Moreover, thery suggested  to avoid a mediastinal drainage from cervicotomy and prefer Uniportal VATS for obtaining a complete evacuation with a mini-invasive approach.

Because DNM is a rare but well-known disease, and treatment strategies are critical to overcome this disease, the topic of this study is important.

There some problem in this present format, and so, I had some comments.

In this study, 18 patients with DNM who underwent surgical treatment were included, and 8 patients had postoperative complications. This resulut is agreeable, however; a multivaliable logistic regression analysis is overfitting statistically.Therefore, the present study design must be limited in its conclusions. 

Author Response

Dear Reviewer, 

I appreciate your comments and I agree with you. 

According with your suggestions, we tried to underly the limits of a retrospective study with small number for driving strong conclusions. 

I hope this version find your endorsement and your agreement. 

Kind regards

Maria Teresa Congedo on the behalf of the co-Authors. 

Reviewer 3 Report

The authors reviewed their single institutional experience of Descending necrotizing mediastinitis (DNM) regarding surgical management. Congratulations to your result.

The following are my thoughts and suggestions.

First, a few case series with analysis retrospectively cannot produce convincing statistical result. I encouraged you to organize relevant studies and rewriting as review article or brief report/short communication if available.

Second, you concluded to avoid a mediastinal drainage from cervicotomy and prefer uniportal VATS for obtaining a complete evacuation with a mini-invasive approach. However, no solid result (only in a few cases operated on according to your experience) drawn from the literature was mentioned and compared. Thus I would not believe this will change the practice in dealing DNM. Thanks.

Author Response

Dear Reviewer,

I appreciate your comments and I agree with you.

According with your suggestions, we completely rewritten the discussion and the conclusion and we  tried to underly the limits of a retrospective study with small number for driving strong conclusions.

I hope this version find your endorsement and your agreement.

Kind regards

Maria Teresa Congedo on the behalf of the co-Authors.
